# Comparing Resampling Algorithms and Classifiers for Modeling Traffic Risk Prediction

**DOI:** 10.3390/ijerph192013693

**Published:** 2022-10-21

**Authors:** Bo Wang, Chi Zhang, Yiik Diew Wong, Lei Hou, Min Zhang, Yujie Xiang

**Affiliations:** 1School of Highway, Chang’an University, Xi’an 710064, China; 2School of Civil and Environmental Engineering, Nanyang Technological University, Singapore 639798, Singapore; 3Engineering Research Center of Highway Infrastructure Digitalization, Ministry of Education, Xi’an 710000, China; 4School of Engineering, STEM College, RMIT University, Melbourne, VIC 3001, Australia; 5College of Transportation Engineering, Chang’an University, Xi’an 710064, China

**Keywords:** traffic crash risk prediction, resampling algorithms, classifiers, performance evaluation measures, feature importance

## Abstract

Road infrastructure has significant effects on road traffic safety and needs further examination. In terms of traffic crash prediction, recent studies have started to develop deep learning classification algorithms. However, given the uncertainty of traffic crashes, predicting the traffic risk potential of different road sections remains a challenge. To bridge this knowledge gap, this study investigated a real-world expressway and collected its traffic crash data between 2013 and 2020. Then, according to the time-spatial density ratio (*Pts*), road sections were assigned into three classes corresponding to low, medium, and high risk levels of traffic. Next, different classifiers were compared that were trained using the transformed and resampled feature data to construct a traffic crash risk prediction model. Last, but not least, partial dependence plots (PDPs) were employed to interpret the results and analyze the importance of individual features describing the geometry, pavement, structure, and weather conditions. The results showed that a variety of data balancing algorithms improved the performance of the classifiers, the ensemble classifier superseded the others in terms of the performance metrics, and the combined SMOTEENN and random forest algorithms improved the classification accuracy the most. In the future, the proposed traffic crash risk prediction method will be tested in more road maintenance and design safety assessment scenarios.

## 1. Introduction

The expressway (which is a class of very high standard road in China) is an essential part of road transportation systems and is also the road type with the highest casualty rate from traffic crashes. According to relevant statistics for China, the annual death toll per 1000 km of expressways is 43.68, which is 4 times that of ordinary roads, and the direct property damage per traffic crash on expressways is US$6500, which is 10 times that of ordinary roads [1]. In particular, the safety level of highways in mountainous areas is worse due to poor alignment conditions [2], high proportion of tunnels [3], complex weather [4], etc. In 2018 alone, there were more than 70,000 crashes in the mountainous areas in China, resulting in more than 100,000 casualties and millions of dollars in losses [5].

Traffic risk evaluation is an important basis for road safety management, which is an important subject in road safety research. Identification of the priorities of road section management through traffic risk evaluation can help designers and managers to efficiently allocate resources and improve road safety [6,7]. The operating environment of an expressway is not affected by pedestrians and personal mobility devices such as bicycles and scooters. There is a high correlation between traffic crashes and road design features [8]. Presently, there are two principal methods for traffic risk evaluation. One method involves the identification of crash-prone sections based on crash data [9]. Sufficient observation time and sample size are the keys to accurate identification of crash-prone road sections. Such an approach entails a certain lag and is hence difficult to incorporate into the design and early stages of roadway operation. The other approach involves the identification of high-risk sections based on road feature analysis [10]. However, though high-risk sections have a greater potential of becoming crash-prone sections, some may not develop into crash-prone sections. Furthermore, this approach only considers individual factors such as road alignment, weather, pavement, etc., and the identified high-risk road sections are often not sufficiently comprehensive. Given the constraints in the afore-mentioned methods, the development of road traffic risk assessment methods that make better use of crash data and with more comprehensive consideration of risk elements is of significant research and application interest.

Newer modeling techniques have been developed in the analysis of motor vehicle crash data, including two recent methods of clustering and classification [11,12,13]. Clustering methods are mainly used to discover potential risk factors and quantify their weight [14]. Traffic risk prediction generally uses decision trees and classifiers. Commonly used classification models include the decision tree classifier [15], rule induction PART [16], lazy classifier [17], Bayes classifier [18], etc. Compared with traditional traffic risk prediction methods, the application of decision tree and classifiers is exploratory rather than inferential. Hereby, more risk factors can be considered, including potential risk factors with unclear impact mechanisms.

The number of road sections at different risk levels typically varies. The spatial distribution of traffic crashes on roads shows different degrees of aggregation given that the occurrence of crashes is affected by various risk sources. Many scholars have taken note of the inherent imbalance in crash datasets, and have proposed balancing methods such as under-sampling, oversampling, and mixed sampling. Using these three methods to deal with imbalanced datasets, Mujalli et al. [19] used the Bayesian classifier to predict the severity of traffic crashes, which mainly considered the number of vehicles, crash mode, speed, lighting, and road conditions. Fiorentini et al. [20] used the random under-sampling of majority class (RUMC) technique to deal with imbalanced datasets to predict the severity of crashes wherein the reliability of the predictions was compared for the four models of random tree, K-nearest neighbor (KNN), logistic regression, and random forest. Danesh et al. [21] applied biogeography-based optimization and invasive weed optimization to deal with imbalanced datasets, and applied the three algorithms of decision tree, support vector machine (SVM), and KNN to construct a prediction model of crash severity. Chen et al. [22] innovatively proposed the ENN-SMOTE-Tomek Link (EST) algorithm to deal with the imbalance problem in the prediction of vehicle spatial sequence features.

At present, classification algorithms are used to predict the severity of traffic collisions with traffic crash datasets with imbalanced characteristics. There is a lack of relevant research on the identification of high-risk road sections [23]. Compared with the prediction of collision severity, the imbalance problem in road section risk prediction is more serious. Data imbalance is not only due to the inherent imbalance of the risk level of road sections but also the existence of potentially high-risk sections that are yet to develop into crash-prone sections [24]. Therefore, new methods need to be proposed to deal with imbalanced crash datasets for road section risk prediction using machine learning algorithms.

In this study, a prediction model of traffic crash risk potential was established. Three types of data-balancing methods (11 algorithms in total) were introduced to deal with traffic-related data. The processed balanced dataset was applied on 7 classifiers and 84 classification models were obtained. Finally, the performance of the established models was compared, and the optimal resampling algorithm and classifier for expressway risk potential prediction were proposed.

The remaining paper is organized as follows: Section 2 presents a brief description of the data used, data-balancing methods, and classifiers. Section 3 presents the findings. Section 4 gives the conclusion.

## 2. Materials and Methods

In this study, traffic crash data and road data on Chinese expressways were collected, which resulted in a dataset that is inherently imbalanced. Unbalanced data can easily lead to the prediction results being biased towards the majority class samples, resulting in a falsely high accuracy of the prediction results [25]. Herein, resampling algorithms and classifiers were applied to improve the negative impact of the class imbalance problem on the model performance. Considering that the potential risk of some roads may be improved by traffic safety facilities, resulting in the predicted road risk level being higher than the actual risk level, a new performance measure was proposed to evaluate the performance of the trained model. Figure 1 shows the procedures employed.

### 2.1. Data

In this study, traffic crash data and road data were collected for the GYX Expressway in southwest China, which were provided by GYX Expressway’s traffic police and expressway operating companies. A total of 4187 traffic crash data records were collected covering a period of 8 years (2013–2020). Crash data included details such as the crash time, crash location, etc. The total length of the two-way four-lane expressway is 210 km. Roadway data included road alignment, pavement condition, tunnels, and expressway facilities. In addition, daily weather data for 8 years (2013–2020) was obtained from the weather website.

The expressway was divided into 420 sections according to the driving direction and kilometers. In order to study the traffic risk of road sections under different weather conditions, four weather conditions were considered: sunny, cloudy, overcast, and rainy. Therefore, there were a total of 1680 sampled units within this study as shown in Table 1. Feature variables (road and weather information) and target variables (crash information) were characterized for each sampled unit.

#### 2.1.1. Data Integrity and Accuracy

Traffic crash data is an important basis for road safety research. The recording of traffic crashes is generally the responsibility of road management companies and traffic police. In China, each expressway has a dedicated agency responsible for managing traffic crashes. Before the advent of mobile phones and monitoring systems, it was difficult to contact the traffic police on the expressway in a timely manner. Drivers would often resolve crashes themselves without going through the traffic police whenever minor traffic crashes occurred. Therefore, in the past, there was a serious lack of traffic crash data for minor crashes. With the popularity of mobile phones and monitoring systems, drivers can now readily contact the traffic police department, resulting in the timely detection of traffic crashes by the police authority. At the same time, traffic crash insurance compensation in China requires the traffic police to issue a liability determination letter. Therefore, practically all traffic crashes on the expressway are recorded by the traffic police department.

The crash data in this study were provided by traffic police and highway management companies, respectively, and included details of the time, location, direction, crash type, and casualties. Herein, the traffic police provided traffic crash data from 2013–2020, with the time and the cause of the crash being recorded. However, the highway management company only provided data for the two years of 2017 and 2018, with recording of the amount of road damage and compensation. The analysis of the crash data entailed two aspects of completeness and accuracy, as follows:

(1). Completeness of crashes: In the comparison of the crash records of 2017 and 2018 provided by the traffic police and highway management companies, no missing incidents were found. In addition, there were many crashes in which vehicles slightly scratched the road guardrails.

(2). Accuracy of crashes: The time of the crash was accurate to the minute. The location information in some crash records in 2013 and 2014 was only accurate to the kilometer level. In the crash records after 2015, the location information was accurate to the 10 m level.

In essence, the crash data used in this study had good completeness and accuracy, thereby laying a good foundation for risk analysis and risk prediction.

#### 2.1.2. Feature Variables

Relevant studies have demonstrated that road alignment, pavement state, tunnels, and expressway facilities are key factors affecting the level of road safety [26,27]. Lee et al. [28] found that factors such as road horizontal alignment, vertical alignment, road surface type, and weather have a significant impact on crash severity. Schloegl et al. [29] used random forests and boosted trees to predict the occurrence of crashes and concluded that the application of feature variables, including road alignment, pavement conditions, traffic volume, and weather variables, can effectively predict the occurrence of traffic crashes.

Altogether, 17 road features and one weather feature were selected as independent variables, with the selected variables being used in the literature and available in our datasets (Table 2).

Among the variables, the maximum curve radius index (*RIMX*) and *RICU* (cumulative radius index) were first proposed in this study, and the calculation formulas are as follows:(1)$$RIMX=MAX(RI_{n})$$
(2)$$RICU=\sum RI_{n}$$
(3)$$RI=\{\begin{array}{c} \frac{R^{*}}{R}\\ 1 \end{array}\begin{array}{c} \left(R<R^{*}\right)\\ \left(R\geq R^{*}\right) \end{array}$$
where RI is an indicator of the radius of the horizontal alignment. The value ranges from 1 to infinity, and the larger the value, the smaller the radius. R is the radius of the horizontal alignment. $R^{*}$ is the minimum radius without superelevation. According to Chinese standards, when the design speed is 80 km/h, $R^{*}$ is 2500 m.

#### 2.1.3. Road Risk Level

In order to quantify the traffic risk of different weather and road sections, this study proposed a new traffic crash statistical index, the temporal-spatial density ratio (Pts), as the target variable. The basic premise of Pts is to characterize the relative safety level of the road section. Considering the spatial influence of the section length on the crash frequency, the number of crashes per kilometer was calculated. Considering the temporal influence of weather on the crash frequency, the number of crashes per day under each weather condition was calculated. Finally, the relative safety level of the road section was characterized by the temporal-spatial density ratio, as defined in Equation (4):(4)$$Pts=\frac{N_{xe}/L_{e}/D_{x}/V_{x}}{N/L/D/V}$$
where Pts is the temporal-spatial density ratio, $N_{xe}$ is the number of crashes with *x* weather in the spatial unit, $L_{e}$ is the length of the spatial unit, $D_{x}$ is the number of days with *x* weather, N is the total number of crashes, L is the total length of the road network, and D is the total number of days.

When Pts = 1, it means that the crash rate of this road section in given weather is at the average level of the road network. When Pts=3, it means that the crash rate of this road section in given weather is 3 times the average level. Herein, the road risk can be divided into three levels: when Pts < 1, the section’ risk level is L_1; when 1 ≤ Pts < 3, the section risk level is L_2; and when Pts ≥ 3, the section risk level is L_3. Pts of 1680 units was enumerated, as shown in Figure 2.

### 2.2. Data Balancing

When there is a large difference in the number of samples among the classes, the problem of class imbalance occurs. A dataset with such a problem is usually called an imbalanced dataset. The imbalance ratio (IR) value is commonly used to measure the degree of imbalance in a dataset. The IR value is the ratio of the number of samples in the majority class to the number of samples in the minority class. In this study, the ratio of road section data for the three risk levels is about 12:2:1, and the IR value is 12, which is an extremely imbalanced dataset.

When using the classifiers, imbalanced data can cause degradation of the performance of the classifier [30]. A popular solution is to balance the number of samples of each category by sampling the original data when the sample size cannot be expanded. The data balancing process can be carried out using three methods: under-sampling, oversampling, and mixed sampling. Under-sampling is used to reduce the majority class samples to achieve equilibrium. Oversampling is used to increase the minority class samples to achieve equilibrium. Mixed sampling combines both methods and can handle both majority and minority class samples. In this study, 11 resampling algorithms were selected from 3 data balancing methods: under-sampling (4), oversampling (5), and mixed sampling (2), as described in Table 3.

### 2.3. Classifier Selection

The classification algorithm is a type of supervised learning. The training of a model with known classes of data can classify unknown data. Classification algorithms are widely used in traffic crash data mining. K-nearest neighbors (KNN), support vector machine (SVM), Bayesian, and Ensemble algorithms are the four most widely used methods [38,39].

#### 2.3.1. KNN Classifier

The KNN classifier performs classification according to the calculated distance, which is the distance from an unlabeled object to all labeled objects [40]. Its advantage is simple, and the application has good adaptability in most scenarios. Since the algorithm relies on the distance for classification, it is necessary to normalize feature variables with large differences in their physical units or scales [41]. However, normalization cannot completely solve the shortcomings of the KNN algorithm’s complete dependence on distance, especially the lack of accuracy in the voting decisions.

According to the number of neighbors, KNN can be divided into fine KNN, medium KNN, and coarse KNN [42]. The number of neighbors in fine KNN is 1, which describes a slight gap between sample classes. Since the value gap of *CULR*, *GAE*, and other feature variables in the data of this study is very small, fine KNN is used for classification. Yigit [43] introduced weight on the basis of distance and proposed weighted KNN, an improved KNN algorithm. Kuang et al. [44] applied the weighted KNN algorithm to improve the accuracy of duration prediction in traffic crashes. Finally, in this study, fine KNN and weighted KNN were selected to represent the KNN method for model training.

#### 2.3.2. SVM Classifier

SVM achieves sample classification by constructing hyperplanes in multi-dimensional space. The distance from the sample point to the hyperplane is called the interval. The interval is the distance from the sample point to the hyperplane. The goal of SVM is to find an optimal hyperplane with the largest interval from each sample [45]. SVM was originally designed to solve the two-class problem, but the present study belongs to the three-class problem. To make the SVM algorithm suitable for this study, the one-versus-rest mode was adopted to train the classifier in two steps: (1) Divide the dataset into two parts: L_1 level and non-L_1 to train the first SVM classifier; (2) using L_2 and L_3 data, train a second SVM classifier.

To train the model with the SVM classifier, a kernel function needs to be used to transform the space for nonlinear data. Commonly used kernel functions are the radial basis function, Gaussian (RBF) kernel, and cubic (polynomial degree 3) kernel, etc. [46]. Finally, in this study, cubic SVM and fine Gaussian SVM were selected to represent the SVM method for model training.

#### 2.3.3. Ensemble Classifier

The ensemble classifier classifies samples by combining multiple models. The basic idea is to learn a set of classifiers and decide the classification result based on their votes. This method has a better predictive performance than the use of a single model. Compared with KNN and SVM, its advantage is that it does not need to normalize the data. There are two commonly used ensemble classifiers: bagging (bootstrap aggregation) and random forest. Bagging classifiers are used to reduce the variance of decision trees [47]. Random forest is an extension to bagging. Each classifier in the ensemble is a decision tree classifier and uses randomly chosen attributes at each node to determine the split. When classifying, each tree votes and returns the most popular class. Malik et al. [48] compared the performance of six classifiers in predicting the severity of road crashes. The study found that random forests, decision trees, and bagging are significantly better than other algorithms in all performance measures. In addition, XGBoost is an emerging algorithm that combines the loss function and the regularization term to build an overall loss function. Shi et al. [49,50] established a driving risk assessment method based on XGBoost, and the risk prediction accuracy rate can reach 89%. On this basis, a risk prediction method for autonomous vehicle decision-making was further established. The prediction accuracy reached 91.7%. Parsa et al. [51] used XGBoost to predict traffic crashes and reported that XGBoost has a more robust effect. Finally, in this study, bagging and random forest were selected to represent the ensemble classifier for model training.

#### 2.3.4. Bayesian Classifier

The Bayesian classifier is a classification method based on Bayes’ theorem. The algorithm decides the classification result by minimizing the probability of misclassification. The naive Bayesian classifier is the simplest and most common classification method in the Bayes classifier [52]. The classification result can be described by an explicit network, and each feature variable is a network node. The naive Bayesian classifier requires input feature variables to be independent of each other [53]. Some feature variables in the horizontal alignment and vertical alignment in this study are not independent of each other. Therefore, the naive Bayes classifier is not suitable for this study.

In order to relax the conditional independence assumption that governs standard naive Bayes learning algorithms, scholars have proposed various solutions [54]. Chen et al. [55] used a Bayesian network to study the causes of vehicle crashes, but this method requires the construction of explicit topological relationships between random variables. Another solution is the AODE algorithm (averaged one-dependence estimators), which does not require explicit topological relationships and improves the classification accuracy at the cost of a modest increase in computation [56]. Finally, in this study, the AODE algorithm was selected to represent the Bayesian classifier training model.

### 2.4. Performance Evaluation Measures

The confusion matrix is an important method used to evaluate the classification effect of a model. The commonly used two-class confusion matrix divides classification results into four categories: true positives (TPs), true negatives (TNs), false positives (FPs), and false negatives (FNs). According to the four results, five measures of accuracy, sensitivity, specificity, precision, and F1 were proposed. This study is a three-classification problem, and the confusion matrix is shown in Figure 3. The equations of commonly used performance measures are as follows:(5)$$\mathrm{Accuracy}:\;Acc=\frac{\sum_{i=j}N_{i,j}}{\sum \sum N_{i,j}}$$
(6)$$\mathrm{Precision}:\;PPV_{n}=\frac{N_{n,n}}{N_{1,n}+N_{2,n}+N_{3,n}}(n=1,2,3)$$
(7)$$\mathrm{Sensitivity}:\;TPR_{n}=\frac{N_{n,n}}{N_{n,1}+N_{n,2}+N_{n,3}}(n=1,2,3)$$
(8)$$\mathrm{Specificity}:\;TNR_{n}=\frac{\sum_{i\neq n}\sum_{j\neq n}N_{i,j}}{\sum \sum N_{i,j}-\sum N_{n,j}}(n=1,2,3)$$
(9)$$F1_{\left(n\right)}=\frac{2\times PPV_{n}\times TPR_{n}}{PPV_{n}+TPR_{n}}(n=1,2,3)$$
where $N_{i,j}$ is the number of road sections, i is the observed risk level of the section, and j is the predicted risk potential level of the section.

When *i* = *j*, $N_{i,j}$ is located on the diagonal of the confusion matrix, which means that the observed risk level of the road section is consistent with the predicted risk level. When *i* < *j*, $N_{i,j}$ is located in the upper right triangle of the matrix, which means that the observed risk level of the road section is lower than the predicted risk level. When *i* > *j*, $N_{i,j}$ is located in the lower left triangle of the matrix, indicating that the observed risk level of the road section is greater than the predicted risk level.

Road crash risk is affected by many factors, which can be divided into 59 risk factors such as road alignment, pavement, environment, presence of work zones, cross-section, and traffic control [57]. This study focused on the impact of inherent properties on road risk. The feature variables considered, except for the road surface state, are all inherent properties of the road. The prediction results characterize the traffic risk potential caused by the inherent properties of the road. The positive impact of safety control measures and other factors on traffic risk was not considered. Meanwhile, traffic crashes have a certain randomness, including the crash location and time. Therefore, there is a significant difference between the number of crashes on high-risk road sections and low-risk road sections under a larger observation time and sample size. When the observed risk level of the section is lower than the predicted risk level, the prediction result is not necessarily inaccurate. For example, some road sections with L_3 traffic risk potential are subject to traffic control measures and their observed risk level is L_2 or L_1. When the observed risk level of the section is greater than the predicted risk level, the prediction result is less affected by the inherent properties of the road, and the risk source may be attributed in part to other factors. Considering the characteristics of this study, a new performance measure, Score, was proposed to evaluate the predicted results, as defined in Equation (10):(10)$$\mathrm{Score}:\;S=\frac{Acc+TPR_{1}+TPR_{2}}{3}$$

## 3. Results and Discussion

In this section, we focus on the following points for the purpose of finding the best method for predicting the traffic risk potential of the road:Analysis of the correlation of 18 kinds of feature variables, retaining the necessary feature variables to construct a dataset.Comparison of the improvement effects of 11 different resampling algorithms on model training and identification of the best resampling algorithm.Comparison of the impact of seven classification algorithms on model performance and identification of the best classification algorithm.Use of the best algorithms and classifiers to analyze the contribution of various feature variables in model training and identification of the key risk factors that affect road risk.

### 3.1. Feature Analysis and Dimensionality Reduction

The focus of feature selection is to select a subset of variables from the input that can efficiently describe the input data while reducing the effects from noise or irrelevant variables and still provide good prediction results [58]. The Pearson correlation coefficient is one of the commonly used indicators in feature selection [59]. The removal of redundant data by correlation analysis can make each input variable relatively independent.

There were 18 features variables in this study, among which tunnel (*TNN*), interchange (*ITC*), service area (*SVA*), and weather (*WET*) were categorical variables and had no direct relationship with the other studied variables. The remaining 14 variables were continuous variables. Using the Pearson correlation coefficient as an indicator, correlation analysis of the continuous variables was carried out, and the analysis results are shown in Figure 4.

It can be seen that the correlation coefficient between the straight length ratio (*STLR*) and curve length ratio (*CULR*) in the horizontal *alignment* variable is −0.75. Considering that a gentle curve has a greater impact on the driver than a straight section, *STLR* was excluded from the feature variables. In addition, the correlation coefficients of the cumulative radius index (*RICU*) with the number of horizontal change points (*HPNE*) and maximum curve radius index (*RIMX*) are 0.84 and 0.82, respectively. So, *HPNE* and *RIMX* were excluded from the feature variables. The correlation coefficient between the average grade (*GAE*) and maximum grade (*GMX*), *GAE* and downhill length ratio (*DWLR*), and *GMX* and *DWLR* in the vertical line shape variable is 0.97, −0.92, and −0.9, respectively. *GAE* can represent *GMX* and *DWLR* to a great extent, so *GMX* and *DWLR* were excluded. Finally, 13 feature variables were retained, including 4 categorical variables and 9 continuous variables.

### 3.2. Imbalanced Versus Balanced Datasets

The original dataset included 1680 road sections, and 80% of the samples were selected as training data (1260 road sections in total). The sample size distribution of the training data was: 1093 L_1 risk sections, 168 L_2 risk sections, and 83 L_3 risk sections. To address the imbalanced dataset problem, under-sampling, oversampling, and mixed balanced datasets were developed using 12 different resampling techniques. The normalization algorithm was used in the establishment of the dataset. Table 4 shows the totals in all used datasets and their distribution among different risk classes. It can be seen that except for the ALLKNN and ENN algorithms, the other algorithms basically achieved the goal of data re-collection. When the ENN algorithm was used to process unbalanced data, the sample size of L_2 risk road sections in the processed dataset was abnormally reduced. However, the algorithm combining SMOTE and ENN (i.e., SMOTEENN) can effectively improve the effect of data re-collection.

### 3.3. Classifier Performance

Based on the 12 datasets in Table 4, we trained the different models using the 7 classifiers described in Section 2.3. We randomly selected 80% of the original dataset as training data and the remaining 20% as the test set. The random assignment process was performed 10 times, with a total of 10 sets of “training + testing” data. Each training data was balanced with 11 resampling techniques. Based on the processed dataset and the original dataset, 7 kinds of classifiers were used for model training, and a total of 84 groups of models (10 in each group) were obtained. In order to verify the validity of the model of the method, we applied the trained model to predict the test data and compare the observed and predicted results. The performance of each group of models was evaluated. The mean and standard deviation of each of the six performance indicators were calculated, as shown in Appendix A (Table A1).

Then, we calculated the scores of all models using Equation (10). All results were ranked according to Score (the average of Score was used first, and when the averages were equal, it was sorted according to the standard deviation), as shown in Appendix B (Table A2). Table 5 lists the top 10 results.

With respect to the results obtained by the testing set, the following findings were extracted:Evaluating the performance of a model based solely on the accuracy, precision, sensitivity, and specificity did not result in a suitable model. For example, the accuracy of the model trained by the SVM algorithm in the original dataset is 81%, the precision of L_1 is 81%, the sensitivity is 100%, and the specificity is 0. However, when the precision and sensitivity of L_2 and L_3 are both 0, the specificity is 100%. Obviously, it cannot meet the needs of road section risk classification.The F1 indicator can integrate the accuracy, precision, sensitivity, and specificity, and collectively evaluate the classification effect of a single category. According to F1_(1)_, it is found that the original dataset and the under-sampling dataset are biased towards the majority class (L_1). According to F1_(2)_ and F1_(3)_, it is found that the muse ensemble classifier (XGBoost or random forest) can effectively improve the classification effect of minority categories (L_2 and L_3).Not all data balancing algorithms improve the classification performance. Meanwhile, whether the data is balanced hardly affects the ranking order of the classifier performance.The ensemble classifier is used in the top 10 method combinations in the Score index. Among them, the model trained by SMOTEENN and random forest has the highest score.

Using Score to evaluate the model performance, the bias of the results is unclear. Although Score can theoretically take into account both the accuracy and sensitivity, there may be two types of classification effects that are excellent while the other is poor. In order to verify the Score evaluation results, the probability distributions of the prediction results of the models trained by random forest and XGBoost were compared, as shown in Figure 5.

The expected result is that the predicted probability of the positive class (the predicted result is consistent with the observed result) is higher than that of the other two negative classes. Similar to the distribution of the observed L_3 of random forest in Figure 6, the predicted probability of L_3 is almost always very high. It shows that there is a certain regularity in the traffic crash risk level at L_3, and the prediction model can be effectively classified. It can be seen that the classification effect of the random forest model is significantly better than that of the XGBoost model. These results are consistent with the trend of the Score values, indicating that the use of Score to evaluate the classification effect is reliable, and can balance the sensitivity of the three risk levels and the overall accuracy.

According to the results of Score in Appendix A, the tree-based ensemble classifiers outperform the KNN and SVM methods in dealing with the three-class problem in this study. The reason may be that KNN and SVM are “distance”-based classification algorithms, which are not applicable to a traffic crash dataset, even if the data are normalized. Both the random forest and XGBoost showed better results in this study. This is consistent with the findings of Schloegl et al. [29] and Parsa et al. [51]. This shows that the ensemble method has a better application prospect in traffic crash-related research.

At the same time, it is found that various resampling methods can improve the classification effect of the random forest and XGBoost. It shows that the choice of classifier is more important than the data balance method. In this study, the model trained by the random forest method based on the SMOTEENN-balanced dataset is the best, which is consistent with the results of Mujalli [19]. This again shows that the mixed resampling method can deal with the imbalance problem in traffic crashes more effectively.

Schloegl [60] studied the impact of environmental factors on the occurrence of traffic crashes and found that XGBoost has a higher classification effect than random forest. This is inconsistent with the analysis results of this study. The reason may lie in the small sample size and the existence of potentially high-risk road sections, resulting in a large proportion of noise in the data. Compared with random forest, XGBoost is prone to overfitting due to noisy samples. However, the effect of XGBoost may improve with an increased number of feature variables and samples.

Finally, SMOTEENN and random forest were selected to build a traffic crash risk prediction model of expressway.

### 3.4. Feature Importance and Interpretation

Random forest is essentially a tree model, and its classification method is nonlinear. Feature importance measures alone do not represent an impact on classification. However, the ranking of feature importance represents its relative influence. Three feature importance indicators were used in this study: Gini importance, permutation importance, and SHAP importance. The Gini importance represents the probability of each feature in the tree node, and the sum of its values is 1. The larger the value, the greater the importance of the feature in the node. However, if there is a correlation between multiple features, the feature that participates in the classification first will have a higher importance, and the feature value that participates in the classification later will be reduced. So, this easily results in an unstable order of influence. Permutation importance is used to analyze the feature importance by disrupting the data of a feature after model training is completed and observing whether the model accuracy is reduced. When the value is less than 0, it means that the observation accuracy is improved after the data is scrambled. However, in this study, the accuracy cannot evaluate the effect of the model, so it cannot truly reflect the feature importance. SHAP (Shapley additive explanation) is proposed based on game theory. The SHAP baseline value of a feature is calculated to indicate its importance, and its value is consistent with the influence weight value of the feature, which can better describe the classification contribution of the feature [61]. Feature importance analysis was carried out based on the trained model, as shown in Figure 6.

In Figure 6, the permutation importance is relatively unstable. For example, the permutation importance of *SRI* is inconsistent with the other two importance indicators. The ranking results of Gini and the SHAP importance are basically the same. Two horizontal alignment feature variables, *RICU* and *STLR*, have high importance. Among the three vertical alignment feature variables of *GAE*, *GDF*, and *VPNE*, *GAE* and *GDF* are more important while *VPNE* is less important. The four road condition indicators of *PCI*, *RDI*, *SRI*, and *RQI* have high importance. The three structural feature indicators of *TNN*, *ITC*, and *SVA* can be ignored except for *TNN*. The results show that the traffic risk potential of this study section is highly correlated with horizontal alignment, vertical alignment, road surface conditions, and weather, and is less affected by expressway facilities.

This study focuses on the prediction of traffic risk based on inherent road properties rather than crash rate prediction. Inherent road properties are only part of the contributions to crash risk. It should be emphasized that the purpose of this study was to optimize road design and road maintenance programs. Therefore, the model in this study does not need a particularly high accuracy, but the influence of feature variables on the prediction results must be consistent with the theory of road risk.

The feature importance analysis results show which feature variables are more important. However, its influence law is not clear, and the influence law of some factors may be inconsistent with the theory.

The influence of inherent road attributes on the traffic risk potential was analyzed, and the reliability of the proposed traffic crash risk prediction method was verified in this study. The partial dependence was introduced to analyze the influence of feature variables on the prediction results. The larger its value, the greater the influence of the feature variables on the results. It is worth noting that *RICU*, *GAE*, and *PCI* are the three most important feature variables, representing the horizontal alignment, vertical alignment, and pavement status, respectively. The partial dependence plot under the interaction of *RICU*, *GAE*, and *PCI* is shown in Figure 7.

According to Figure 7, we can find the following points:When *RICU* is greater than 30 and *GAE* is less than 0, the partial dependence value of the road section with the traffic risk level L_3 is greater than 0.32. At the same time, when *RICU* is less than 5 and *GAE* is greater than 0.02, the partial dependence value of the road section with the traffic risk level L_1 is greater than 0.65. This shows that the model established in this paper can effectively identify two types of typical high-risk road sections: downhill small radius curves or downhill continuous curved road sections.When *RICU* is greater than 20 and *PCI* is less than 98, the partial dependence value of the road section with the traffic risk level L_3 is greater than 0.32. However, when *RICU* is less than 5 and *PCI* is less than 98, the partial dependence value of the road section with the traffic risk level L_1 is greater than 0.64. This shows that in the model established in this paper, the pavement condition has a greater impact on the driving safety of small-radius road sections or continuous curved road sections while it has a lesser impact on the driving safety of straight sections.When *GAE* is less than 0 and *PCI* is less than 95, the partial dependence value of the road section with the traffic risk level L_3 is greater than 0.30. However, when *GAE* is greater than 0.02 and *PCI* is greater than 0.4, the partial dependence value of the road section with the traffic risk level L_1 is greater than 0.5. This is basically consistent with the above two rules. At the same time, this also shows that in the model established in this paper, the downhill section has higher requirements regarding the pavement performance, and the higher the uphill slope, the safer the road section.

The results show that the larger the *RICU* (the smaller the radius of the curve or the higher the number of curves), the higher the risk level of the road section. The smaller the *GAE* (downhill and the greater the grade), the higher the risk level of the road section. The smaller the *PCI* (more severe road damage), the higher the risk level of the road section. The classification principle of the model is consistent with the traffic risk mechanism and the results of Elvik [62]. This shows that the traffic crash risk prediction model is reliable.

Compared to the study by Schlögl M [29], the variables in this study are very simple in terms of weather. However, the results also show that weather is an important variable after the road alignment and pavement condition. However, the impact of tunnels and expressway facilities on traffic crashes seems to be very low, which may be due to three aspects. First, the data sample size was small and there are only nine interchanges and two service areas in this study. Second, this study excluded the crash data of ramps, and some interchange entrance and exit crashes may be recorded as ramp crashes. Third, the service area itself has a low impact on traffic crashes.

## 4. Conclusions

For highway management to develop precise and effective solutions, the traffic risk potential must be predicted accurately. The inherent properties of the road are the main factors affecting the traffic risk potential, which require many variables to be described. Machine learning techniques offer an efficient method for analyzing many variables at once, which is difficult to achieve with traditional data analysis techniques. Therefore, by evaluating the performance of several resampling approaches and classifiers, this research developed a new model for predicting the risk potential of highway traffic. The main conclusions are as follows:In predicting the risk potential of expressway, horizontal alignment, vertical alignment, pavement, and weather play a significant role. This study provided 18 feature variables from 5 different perspectives, and 13 feature variables were retained after dimensionality reduction. Incorporating Gini importance, permutation importance, and SHAP importance, three feature importance indicators, the contribution of the feature variables to the model was ranked. The results showed that 11 of these feature variables have a high contribution to the prediction of the risk potential.The ensemble classifier demonstrated a good performance in processing traffic accident data, and the addition of a resampling algorithm further enhanced the classifier’s performance. By comparing the prediction results of XGBoost and the random forest algorithm, it was demonstrated that the model performance index Score provided in this research can effectively assess the performance of the three-level traffic risk prediction model.The combination of SMOTEENN and the random forest algorithm developed the best model for predicting the highway traffic risk potential. The three most representative feature variables, *RICU*, *GAE*, and *PCI*, were analyzed using the partial dependence plot, and the results showed that the classification principle of the established traffic crash risk prediction model was consistent with the objective risk effect law.

This research demonstrated the effectiveness of employing machine learning algorithms to assess traffic crash risk. In addition, it provided recommendations for data balancing and classification algorithm selection for machine learning applications based on the inherent features of roads. This research proposed a method for traffic crash risk prediction, which may be utilized to enhance road design and maintenance planning.

In the future, the sample size and feature variables of the dataset will be further expanded for a more comprehensive analysis of the impact on road crash risk. At the same time, an interpretable classifier will be applied in depth to study the mechanism of inherent road features and traffic risk.

## Figures and Tables

**Figure 1 ijerph-19-13693-f001:**
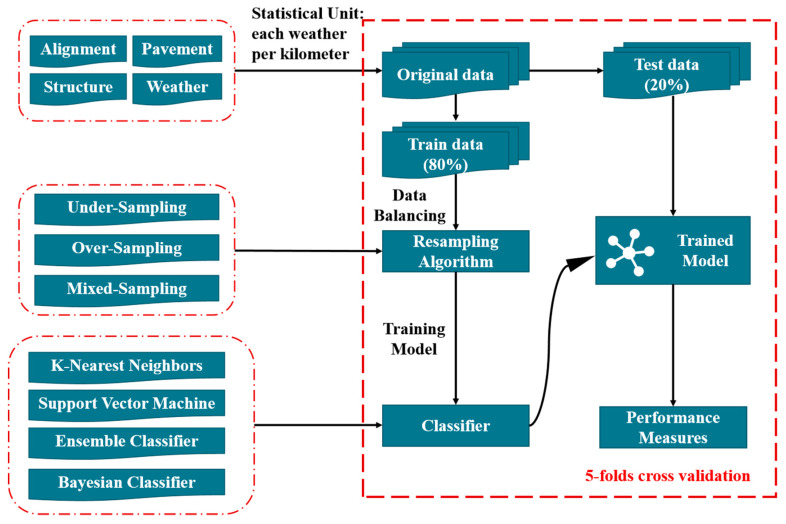
Section division by the homogeneous method; stratified k-fold cross-validation was used to compare the developed models, which was repeated 5 times.

**Figure 2 ijerph-19-13693-f002:**
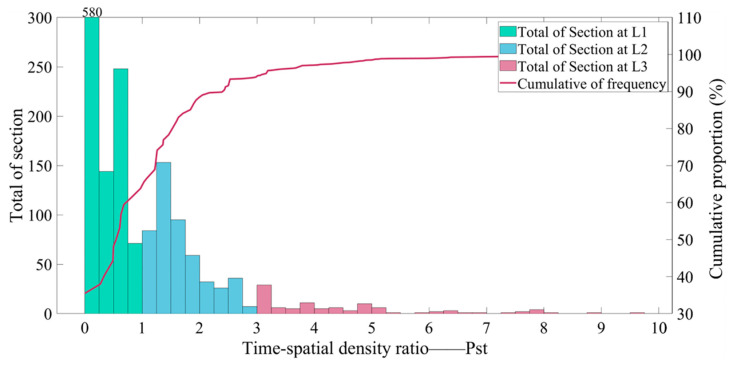
Statistics on the number and cumulative frequency of sections at differentiated risk levels; each 1 km section is counted 4 times according to the weather conditions (sunny, cloudy, overcast, and rainy).

**Figure 3 ijerph-19-13693-f003:**
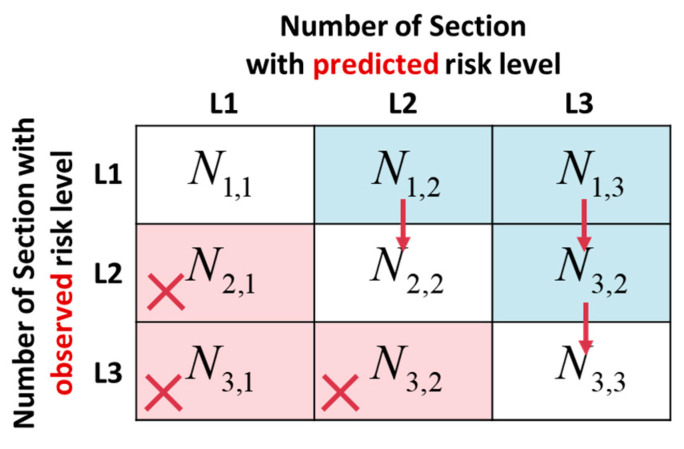
Three-category confusion matrix in this study.

**Figure 4 ijerph-19-13693-f004:**
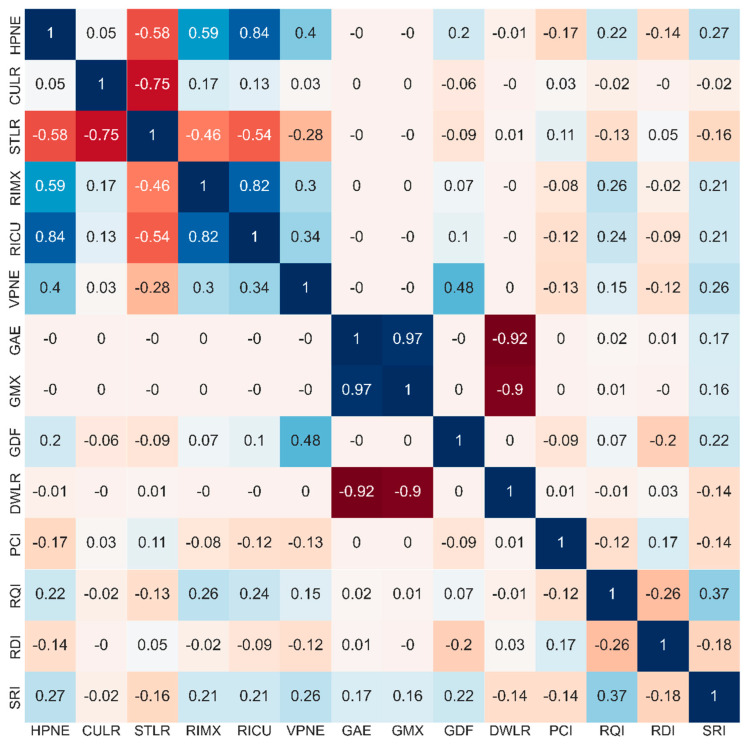
Pearson correlation matrix of continuous feature variables. If the correlation coefficient is greater than 0, it is a positive correlation, and if it is less than 0, it is a negative correlation. The stronger the correlation, the darker the color. The stronger the positive correlation, the closer to blue; the stronger the negative correlation, the closer to red.

**Figure 5 ijerph-19-13693-f005:**
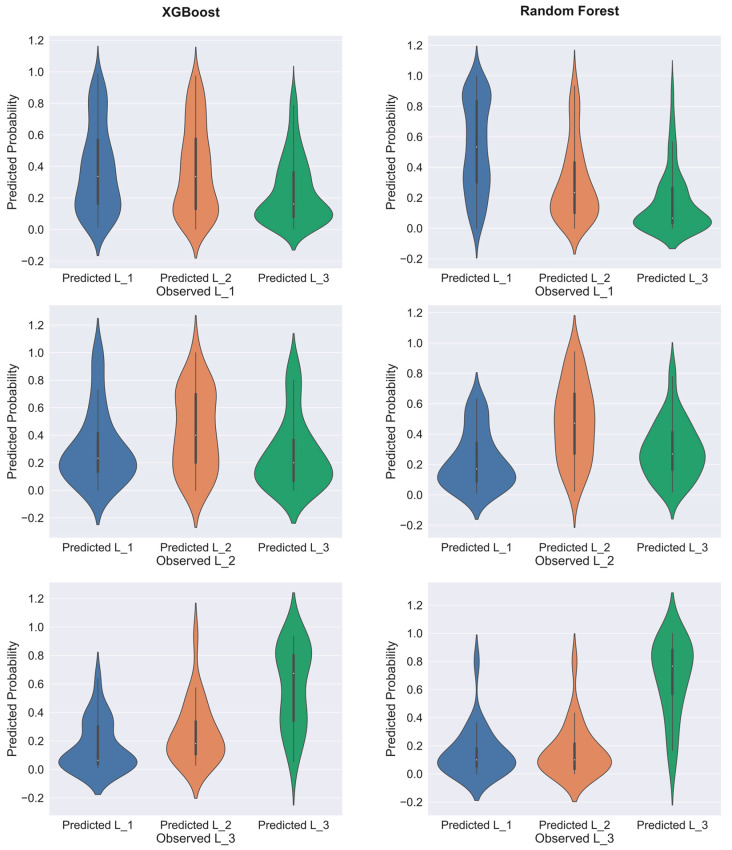
Predicted probability distribution of models trained by random forest and XGBoost. The test data are the same, and the training data are processed by different balancing algorithms: random forest uses SMOTEENN-balanced data and XGBoost uses RUS-balanced data.

**Figure 6 ijerph-19-13693-f006:**
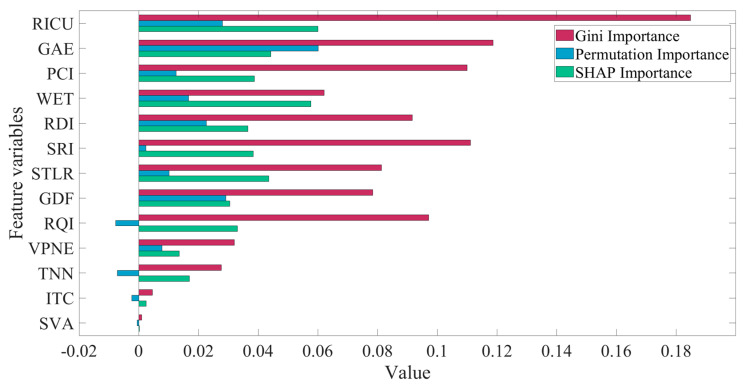
Feature importance (Gini importance, permutation importance, SHAP importance), sorted in accordance with the combined results of the three importance indicators.

**Figure 7 ijerph-19-13693-f007:**
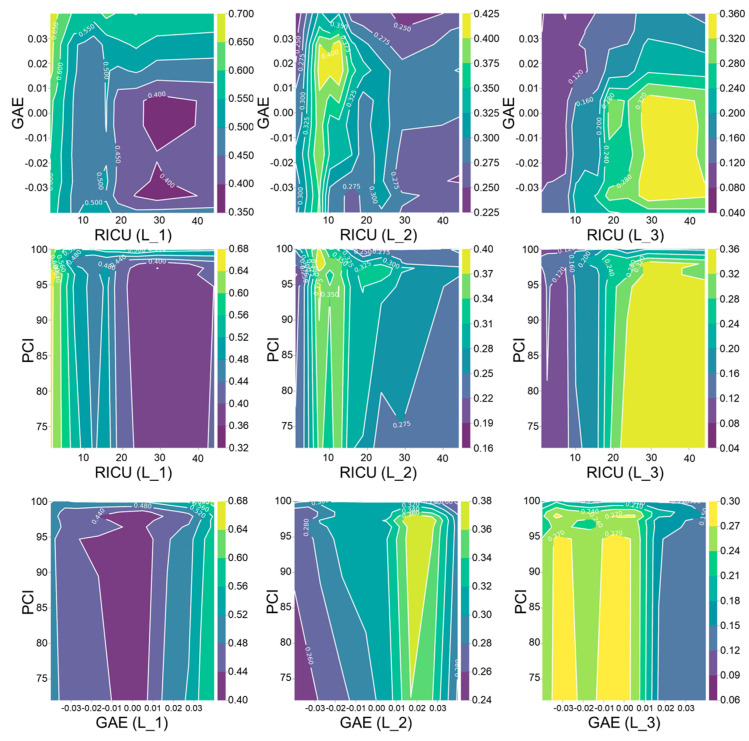
Feature variable partial dependence plot illustrating the effects of section risk by the interaction of *RICU*, *GAE*, and *PCI*; the missing part of the value is filled by interpolation.

**Table 1 ijerph-19-13693-t001:** Overview of the sampled units used in this study.

Number	Stake Number	Driving Direction	Weather
1	K1~K2	forward	sunny
2	K1~K2	forward	cloudy
……			
1679	K209~K210	reverse	overcast
1680	K209~K210	reverse	rainy

**Table 2 ijerph-19-13693-t002:** Overview of the feature variables used in this study.

Factor	Code	Variable	Description
Horizontal alignment	*HPNE*	Number of horizontal change points	The number of change points of horizontal alignment
	*CULR*	Curve length ratio	Ratio of curve length per kilometer (value between 0 and 1)
	*STLR*	Straight length ratio	Ratio of straight length per kilometer (value between 0 and 1)
	*RIMX*	Maximum curve radius index	Maximum value of the inverse of the radius. *RIMX* takes 0 in the straight sections
	*RICU*	Cumulative radius index	Cumulative value of the inverse of the radius
Vertical alignment	*VPNE*	Number of vertical change points	The number of change points of vertical alignment
	*GAE*	Average grade	Average grade per kilometer
	*GMX*	Maximum grade	Maximum grade per kilometer
	*GDF*	Grade difference	The difference in the maximum grade per kilometer
	*DWLR*	Downhill length ratio	The proportion of downhill in the section
Pavement	*PCI*	Pavement surface condition index	The value is between 0 and 100; the lower the value, the more serious the road damage.
	*RQI*	Riding quality index	The value is between 0 and 100; the lower the value, the lower the smoothness of the road surface.
	*RDI*	Rutting depth index	The value is between 0 and 100; the lower the value, the deeper the road rut.
	*SRI*	Skidding resistance index	The value is between 0 and 100; the lower the value, the lower the anti-skid performance of the pavement.
Tunnels and expressway facilities	*TNN*	Tunnel	0—No tunnel in the section; 1—There are tunnels in the section.
	*ITC*	Interchange	0—There is no interchange in the section; 1—There are interchanges in the section.
	*SVA*	Service area	0—There is no service area in the section; 1—There are service areas in the section.
Weather	*WET*	Weather	1—Sunny; 2—Cloudy; 3—Overcast; 4—Rainy.

**Table 3 ijerph-19-13693-t003:** Overview of the 11 resampling algorithms used in this study.

Method	Algorithm	Description
Under-sampling	●Prototype generation (PG)	Generate new samples based on original samples to achieve sample balance. Use K-means to cluster majority class samples and then use cluster centroids as newly generated replacement samples [31].
	●Random under-sampling (RUS)	Some samples are randomly removed from the majority class, so that the samples of each class are balanced.
	●Edited nearest neighbor (ENN)	Apply the nearest-neighbors algorithm to edit the dataset to remove samples with an insufficient neighborhood [32].
	●All-KNN (ALLKNN)	Apply ENN several times and vary the number of nearest neighbors [32].
Oversampling	●Naive random over-sampling (ROS)	Using the method of extraction with replacement, random sampling from minority class samples to replace the existing sample set; can increase the weight of minority class samples.
	●Synthetic minority oversampling technique (SMOTE)	For each minority class sample, the nearest k minority class samples are identified, a sample point is randomly selected each time, the corresponding adjacent sample point is randomly selected, and a new sample point is obtained by interpolating the sample point and adjacent sample point, thereby increasing the minority class samples to balance the data [33].
	●Borderline-SMOTE	This is an improved algorithm of SMOTE. Divide the minority class sample points into “noise points”, “dangerous points”, and “safe points”, and only use the dangerous points when calculating the nearest k minority class samples [34].
	●SMOTENC	This is an improved algorithm of SMOTE. Categorical variables are not properly distanced and interpolated. SMOTENC uses the value difference metric (VDM) algorithm to calculate the distance of categorical variables, which enables the processing of categorical variables [33].
	●ADASYN	Similar to SMOTE, it is also based on k adjacent and interpolation algorithms, the difference being that ADASYN considers other types of samples when calculating k adjacent samples [35].
Mixed sampling	●SMOTEENN	The SMOTE method is used to generate new minority class samples. There may be some noisy samples in the new samples. Apply the ENN method to remove noisy samples and obtain cleaner data [36].
	●SMOT-Tomek Links	Similar to SMOTEENN, Tomek Links are applied to remove noisy samples to obtain cleaner data [37].

**Table 4 ijerph-19-13693-t004:** Number of sections with different risk levels in the different datasets.

Dataset	Balancing Algorithm	Total	L_1	L_2	L_3
Original dataset	/	1344	1093	168	83
Under-sampling	●PG	249	83	83	83
	RUS	249	83	83	83
	ENN	1124	1023	18	83
	●ALLKNN	1065	948	34	83
Over-sampling	ROS	3279	1093	1093	1093
	SMOTE	3279	1093	1093	1093
	ADASYN	3278	1093	1099	1086
	Borderline-SMOTE	3279	1093	1093	1093
	SMOTENC	3279	1093	1093	1093
Mixed sampling	SMOTEENN	2394	598	798	998
	SMOT-Tomek Links	3219	1068	1066	1085

**Table 5 ijerph-19-13693-t005:** Top 10 models ranked according to Score; when the average value of Score was similar, S.D. was used.

Rank	Resampling Algorithm	Classifier	Score	
			Mean	S.D.
1	SMOTEENN	Random forest	0.51	0.04
2	RUS	Random forest	0.47	0.04
3	RUS	XGBoost	0.46	0.04
4	ROS	Random forest	0.46	0.04
5	SMOTENC	Random forest	0.45	0..03
6	ROS	XGBoost	0.45	0.04
7	SMOTE	Random forest	0.44	0.03
9	SMOT-Tomek Links	Random forest	0.44	0.03
8	ADASYN	Random Forest	0.44	0.04
10	SMOTEENN	XGBoost	0.44	0.04

## Data Availability

The datasets used and/or analyzed during the current study are available from the corresponding author on reasonable request.

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
