# Peer review of "Comparing Resampling Algorithms and Classifiers for Modeling Traffic Risk Prediction"

_ijerph, 2022, doi:10.3390/ijerph192013693_

Round 1
Reviewer 1 Report
The following minor corrections are needed.
- What is the difference b/n 2.1.1. Data Integrity and Accuracy and 2.1.2. Data Integrity and Accuracy in p. 4??
- Row 450 and 451 in p. 14: Among the three longitudinal linear feature variables of GAE, GMX and VPNE, GAE and GMX are more important, while VPNE is less important. How do you come to this conclusion? GMX is not given on Fig. 5 as feature variable! GMX and DWLR were excluded since GAE can represent GMX and DWLR to a great extent.
Reviewer 2 Report
This paper did an interesting study on predicting traffic risk with inherent attributes and machine learning modeling. The risk index is based on crash data and 18 variables are proposed to do model classification. Overall, this paper proposed a well-performed method to predict traffic risk. The structure of this paper, however, might need improvement. My suggestions are as below:
-
The structure is not very clear. Section 4, Discussion, might be better if included in the method and/or results.
-
The conclusion is not concise. Please summarize your findings.
-
Although test data is used to validate the accuracy of the model, I’d also like to suggest using some external datasets to validate your model.
Minor revision and accept.
Reviewer 3 Report
This paper analyzes the issue of road traffic crash prediction, using many research methods and selecting many road characteristic variables, and the paper is rich in research content. However, in general, the research of the paper needs to be further focused and the results need to be further analyzed and interpreted, with the following specific suggestions:
1. “road section” or “road segment”? Please choose the appropriate wording and be consistent throughout the paper.
2. Road characteristics are time-invariant variables, which certainly have an impact on road traffic crashes, but the time span of the data in this paper is 8 years, in which the means of traffic management and regulations are getting better and better, these time-invariant variables also have an impact on traffic safety, how to consider the impact of these factors? How does the model constructed in this paper interpret the influence of these factors?
3. What is the impact on the modeling and model results if a balanced transformation is forced on the paper, given the high imbalance of the data (and the IR value is 12, which is an extremely imbalanced dataset, p6 line 213-214)? Has this been considered and evaluated?
4. Is it reasonable to select variables by their correlation? (p11 line 360-371)
5. Suggest that Figure 7 be optimized or that only representative data be selected for illustration, or that Figure 7 be deleted or placed in the appendix.
6. Suggesting more analysis and discussion of the results of Figure 8, which currently has only a brief description of the results.
7. This paper uses many methods (11 resampling algorithms, 4 classifier methods) and many feature variables, and tends to the selection of methods in the final conclusion, but does not show and interpret how the selected features affect traffic crashes in depth, thus feeling that the research of this paper is not focused enough. It is recommended that the authors focus on the method selection or the influence of the road characteristics on traffic crashes in this paper.
8. It is recommended to check the references, keep the format in line with the journal norms and have consistency, and add the volume, issue, page number/article number of the references.
9. The keywords are not representative enough for the paper and it is recommended to optimize them.
Round 2
Reviewer 3 Report
Thanks to the authors for their efforts to revise the paper. It is recommended that the title of the paper can be optimized accordingly, as the current title is not relevant enough to the main content of the paper. Other than that, I have no further comments.
Author Response
Dear Reviewer,
We are grateful that you offered us an opportunity to revise the manuscript "Modeling Traffic Crash Risk of An Expressway with Correction for Imbalanced Data" (ID: ijerph-1961508). We appreciate the time and effort that you spent on the manuscript. The comments are very helpful for us to improve our manuscript. Our replies are provided below in bold.
- It is recommended that the title of the paper can be optimized accordingly, as the current title is not relevant enough to the main content of the paper.
Answer:Thank you very much for your suggestion. We have changed the title of the article to "Comparing Resampling Algorithms and Classifiers for Modeling Traffic Risk Prediction".
Thank you for reviewing our manuscript . This has greatly improved the quality and expression of this manuscript. We hope we have properly addressed all your concerns and the manuscript is now duly revised. If not, we are glad to receive any further feedback from you. Thank you.